# ZIKA Virus infection in pregnant women in French Guiana: More precarious-more at risk

**Edouard Hallet[1], Claude Flamand[2], Dominique Rousset[3], Timothée Bonifay[1], Camille Fritzell[2], Séverine Matheus[3], Maryvonne Dueymes[4], Balthazar Ntab[5], Mathieu Nacher[1,6]***

**1** DFR Santé, Université de Guyane, French Guiana, **2** Epidemiology unit, Institut Pasteur in French Guiana, Cayenne, French Guiana, **3** Arbovirus National Reference Center, Institut Pasteur in French Guiana, Cayenne, French Guiana, **4** Laboratory, Centre Hospitalier Andrée Rosemon, Cayenne, French Guiana, **5** Medical information Department, Centre Hospitalier de l'Ouest Guyanais, Saint-Laurent du Maroni, French Guiana, **6** Centre d'Investigation Clinique Antilles Guyane, INSERM 1424, Centre Hospitalier de Cayenne, Cayenne, French Guiana

* mathieu.nacher@ch-cayenne.fr

## Abstract

### Background

A recent study in French Guiana suggested that populations living in precarious neighborhoods were more at risk for Chikungunya CHIKV than those living in more privileged areas. The objective of the present study was to test the hypothesis that Zika virus (ZIKV) infection was more frequent in precarious pregnant women than in non-precarious pregnant women, as reflected by their health insurance status.

### Methods

A multicentric cross-sectional study was conducted in Cayenne hospital including ZIKV pregnant women with serological or molecular proof of ZIKV during their pregnancy between January and December 2016. Health insurance information was recorded at delivery, which allowed separating women in: undocumented foreigners, precarious but with residence permit, and non-precarious.

### Results

A total of 6654 women were included. Among them 1509 (22,7%) had confirmed ZIKV infection. Most women were precarious (2275/3439) but the proportion of precarious women was significantly greater in ZIKV-confirmed 728/906 (80.4%) than the ZIKV-negatives 1747/2533 (69.0%), p<0.0001. There were 1142 women classified as non-precarious, 1671 were precarious legal residents, and 1435 were precarious and undocumented. Precariousness and undocumented status were associated with a higher prevalence of ZIKV during pregnancy (adjusted prevalence ratio = 1.59 (95%CI = 1.29–1.97), p<0.0001), (adjusted prevalence ratio = 1.5 (95%CI = 1.2–1.8), p<0.0001), respectively.

**Data Availability Statement:** In France, all computer data (including databases, in particular patient data) are protected by the National Commission on Informatics and Liberty (CNIL), the

national data protection authority for France. CNIL is an independent French administrative regulatory body whose mission is to ensure that data privacy law is applied to the collection, storage, and use of personal data. As the database of this study was authorized by the CNIL, we cannot make available data without prior agreement of the CNIL. The data may be made available by the authors but according to French law, researchers wishing to obtain the data must obtain additional authorization with the CNIL. In practice, the first step for any researcher wishing to obtain the data should be to ask the Coordination Regionale de lutte contre le SIDA (corevih@ch-cayenne.fr) who will guide the researcher through the process with the CNIL.

**Funding:** The authors received no specific funding for this work.

**Competing interests:** The authors have declared that no competing interests exist.

## Conclusions

These results illustrate that in French Guiana ZIKV transmission disproportionately affected the socially vulnerable pregnant women, presumably because of poorer housing conditions, and lack of vector control measures in poor neighborhoods.

### Author summary

A recent study in French Guiana suggested that populations living in precarious neighborhoods were more at risk for chikungunya CHIKV than those living in more privileged areas. The objective of the present study was to test the hypothesis that Zika virus (ZIKV) infection was more frequent in precarious pregnant women than in non-precarious pregnant women as reflected by their health insurance status. A multicenter cross-sectional study was conducted including ZIKV pregnant women with serological or molecular proof of ZIKV during their pregnancy between January and December 2016. Health insurance information was recorded at delivery, which allowed separating women into: undocumented foreigners, precarious but with residence permit, and non-precarious. Overall 6654 women were included. Among them, 1509 (22,7%) had confirmed ZIKV infection. The majority of women were precarious, but the proportion of precarious women was significantly greater in ZIKV-confirmed 728/906 (80.4%) than the ZIKV-negatives 1747/2533 (69.0%). Precariousness and undocumented status were associated with a higher prevalence of ZIKV acquisition during pregnancy. The present results illustrate that in French Guiana, as elsewhere, ZIKV transmission disproportionately affected the socially vulnerable pregnant women, presumably because of poorer housing conditions, and lack of vector control measures in poor neighborhoods.

## Introduction

French Guiana (FG) is a French overseas territory in South America. It has the highest Gross Domestic Product per capita in South America, and thus attracts numerous migrants in search of better economic opportunities. Women, and notably migrant women in FG, are especially concerned by inequity with low school enrolment, unemployment, teen pregnancies, and single parenthood [1]. The increase of vulnerable populations is a challenge for the health and social systems on a scale that is far greater in French Guiana than in mainland France[2]. In studies on health inequalities in French Guiana, being an immigrant is also associated with poverty, vulnerability and difficulties in accessing care leading to poor health outcomes.

Poor neighborhoods, with informal housing, lack of sanitation and presence of vector breeding places are especially suited for *Aedes aegypti*, the main vector for arboviruses in FG. This urban mosquito preferentially breeds around human dwellings, in outdoor water storage containers and in any recipient containing stagnant rain water. Densely populated areas with sustained human activity during the day are perfect for a daytime feeder that can bite several people in a short period of time [3].

The autochthonous transmission for the first chikunguya (CHIKV) outbreak in FG (2014), mainly affected foreign populations with precarious social status and those living in poor neighborhoods. In contrast, after further epidemics of dengue virus (DENV) in 2013, populations living in richer areas were mostly concerned. One hypothesis was that poor populations were more likely to have been immunized during previous DENV epidemics because of

greater vector densities and transmission in these areas, whereas in rich neighborhoods populations were more likely to come from France and not be immunized thus being at risk for clinical disease[4].

The distribution of microcephaly in Brazil suggested a possible link between poverty and the risk of Zika virus -related microcephaly[5]. This epidemic spread to neighboring French Guiana in 2016, but until now there was no study of the impact of poverty on the prevalence of infection. In order to deal with the important proportion of asymptomatic Zika virus (ZIKV) infections [6] and a risk of delayed access to medical care in poor populations,[7,8] ZIKV serology was recommended at each trimester for all seronegative pregnant women. A first analysis of data collected during the 4 first months of the outbreak provided a representative picture of the spectrum of disease of ZIKV infection in pregnant women and mapped the intensity of the epidemic throughout French Guiana[6], however it did not look at specific subpopulations of pregnant women. However, the serological surveillance at each trimester during the ZIKV outbreak in 2016 allowed us to test whether the poorest pregnant women were more likely to acquire ZIKV than more socioeconomically privileged women. The objective of the study was thus to compare the prevalence of ZIKV contact in FG for pregnant women according to their health insurance status (HIS).

## Materials and methods

A multicenter, cross-sectional study was conducted in Cayenne (CHAR), Kourou (CHK) and Saint Laurent (CHOG) hospital, the main cities and the main hospitals in FG.

### Population

We included all deliveries, spontaneous abortions or pregnancy terminations for medical reasons between January 2016 and December 2016, 3 months after the official declaration of the end of the outbreak. Women without ZIKV serology were excluded. We also included women with positive RT-PCR even if they did not have any serology. Fig 1 shows the study flowchart. The STROBE checklist is indicated in S1 Checklist.

### Data collection

The dataset was centralized by the national reference center for arboviruses (CNR arbovirus) at Pasteur institute in Cayenne. Data were collected by clinicians, pediatric nurses, or midwives in charge of pregnancy monitoring in the 3 hospital maternity wards. clinical, socio-demographic and geographical individual data were collected at enrolment and at each obstetrical consultation by interviewing the pregnant women. The following variables were available: age, area of residence, pregnancy trimester and pregnancy outcomes (type, date, attendant, place, and last menstruation date). Health insurance information was obtained from the hospital information systems (PMSI) of Cayenne and Saint Laurent du Maroni hospital, but not from Kourou hospital.

### Health insurance status

For health insurance status, we distinguished between patients in a precarious social situation and those who were not. Patients without any health insurance[9], those benefiting from free universal health care called Protection Universelle Maladie (PUMa) (which allows access to health for legal residents who are not already covered usually associated with a complementary insurance the CMUc), or those benefiting from "state medical aid" or "AME" (government run insurance program specifically conceived for undocumented migrants who become

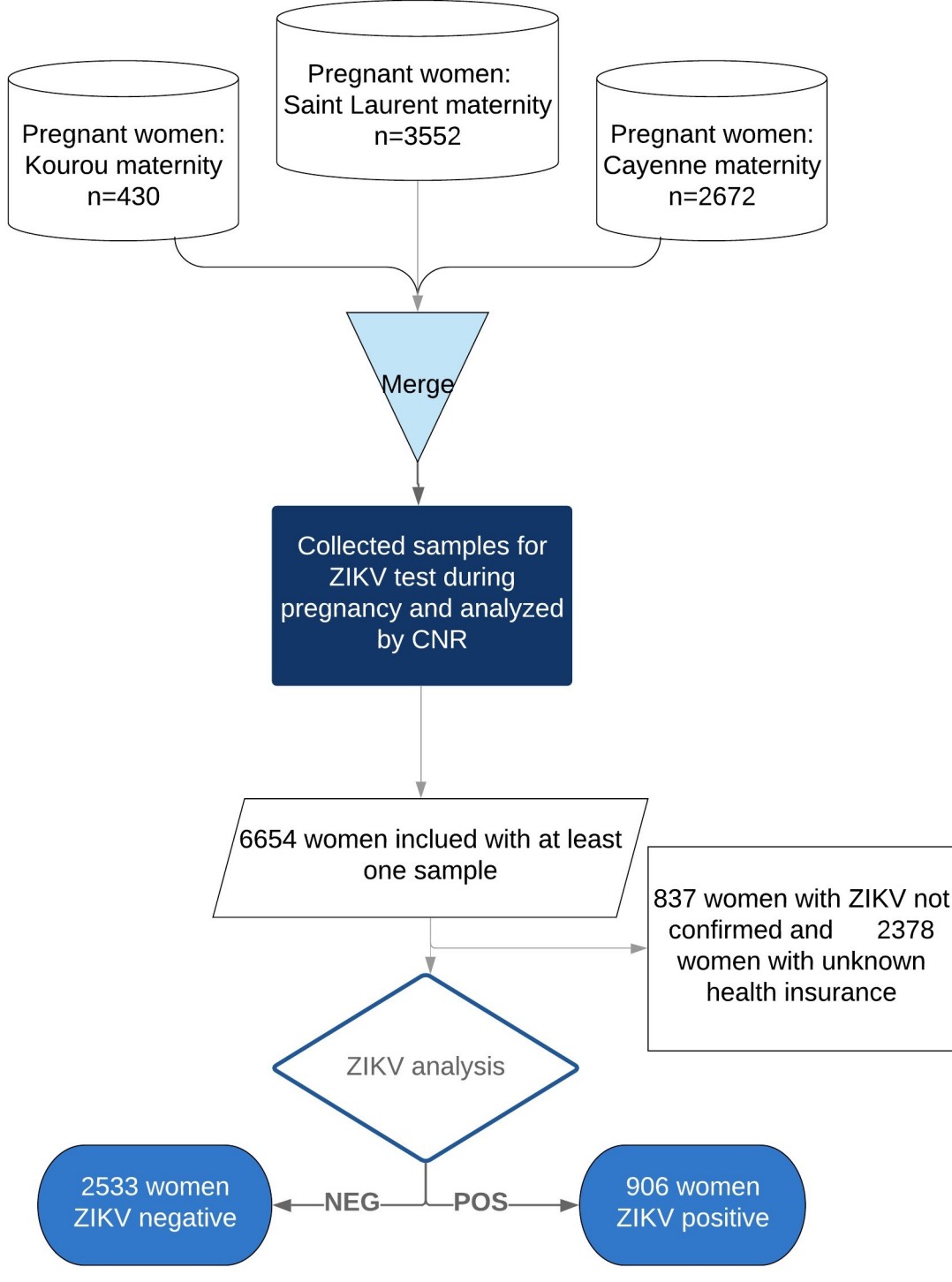

**Fig 1. Study flowchart.** Origin of patients and sample selection.

eligible after 3 months of residency in a French territory) were considered to be in a precarious social situation. Indeed, the annual income of a single person should be less than 8653.16 € (a very low income for French Guiana) to benefit from the above health insurance regimens. Persons with regular social security were considered non precarious. Women seen in Kourou

maternity had no health insurance information. Women without health insurance can receive free care (including ZIKV screening) at the hospital after receiving a "PASS" voucher from a hospital social worker, which allows covering all expenses.

## Study area

We will subsequently refer to different town groupings: Western, savanas, central coastal, and Eastern. The main groups living in the western area are the Maroons, descendants of escaped slaves (15% of the total population of French Guiana) and Amerindians (3%) who live in the south west. Savannas include a mix of previous ethnicities and international workers for aerospace industries. The population of the Central Coastal area is marked by local ethnic groups including Creoles (60% of the total population), people of European ancestry (14%) who essentially come from mainland France and various immigrants from Brazil, Haiti, Caribbean islands, China and southern Asia. The eastern region includes a mix of creole and Amerindian populations.

## Laboratory diagnosis

Serum and urine samples collected for RT-PCR analysis were obtained during trimestrial surveillance and during possible acute symptomatic illnesses, or in the presence of fetal death or fetal structural abnormalities. Placenta and amniotic liquid were assayed for ZIKV RNA by real time RT-PCR using the Lanciotti method[10], the RealStar ZIKV RT-PCR kit, or ELISA serology. Analyses were realized by the National Reference Center of Arboviruses of Pasteur Institute in French Guiana and by the Cayenne Hospital laboratory. Neutralization for IgG could not be implemented, therefore we excluded probable cases with negative IgM and positive IgG.

## Statistical analyses

Bivariate analysis of categorical variables used Poisson regression to obtain prevalence ratios rather than odds ratios, which may overestimate the magnitude of association. Available variables were used in the multivariate analysis using modified Poisson regression in order to obtain prevalence ratios. Stata 12.0 was used for analysis (College Station, Texas, USA).

The retrospective use of anonymized data from patient records is authorized by French law. More specifically, in the emergency context of the Zika epidemic it was part of the health authorities' epidemiologic surveillance research efforts aiming to better understand this emerging problem.

## Results

A total of 6,654 pregnant women were eligible for the study between 01/2016 and 12/2016 as pregnant women in FG. Because of the absence of reply from the medical information department in Kourou, the 2,621 women (39%) with unknown health insurance mostly lived in Kourou (20%). For this study, 4,380 pregnant women were included with at least one sample and with available health insurance information reporting. The final population analyzed count 3,439 women, because 837/6654 women had positive IgG and negative IgM serology without immunocapture. Overall 16.5% of precarious women had positive IgG and negative IgM serology without immunocapture whereas 9.82% of women with normal health insurance had positive IgG and negative IgM serology without immunocapture, P<0.001.

The average age of the study population was 27.9 years-old (range 12–52 years). Overall, 2.1 samples/women were collected on average (range: 1–13), with an average time of 78 days± 53

days between two consecutive serologies for the 3,025 women with more than one sample. Women with a unique sample at the end of pregnancy were 1,168 (13.9%). Abortion represented 216 (3.7%) of the 5817 recorded pregnancy outcomes. The term at diagnosis was compiled for 2,968 (86.3%) women with an average term at sample of 28±10 weeks of amenorrhea (range: 2–42 WA). Regarding residence location: Central Coastal area (6 communes) represented 33.8% of the study population (1,163/3,439 births), Western French Guiana (8 communes), represented 62.4% of births (2145/3439), Eastern French Guiana (4 communes), 1.3% of births (45/3439), Savannas (4 communes), 1.3% of births (45/3439), and visitors, 1.2% (41/3439).

Table 1 shows the characteristics of precarious women: younger women, those from western French Guiana were more precarious (P<0.001), and precarious women seemed less likely to be diagnosed in the first trimester.

Among the pregnant women surveyed, 906 (26.3%) had been exposed to ZIKV. Most pregnant women were precarious 2475/3439. The proportion of precarious women was significantly greater in ZIKV confirmed 728/906 (80.4%) than the ZIKV negatives 1747/2533 (69.0%), p<0.0001. Among precarious, irregular immigrants seemed more exposed 97/300 (32.3%) than French nationals 361/ 1310 (27.6%) (p = 0.10).

Precariousness for legal residents (adjusted prevalence ratio APR = 1.37) and undocumented residents (AME: APR = 1.47 and no insurance: APR = 1.59) were associated with a significant increase of ZIKV prevalence adjusted for residence location (Table 2).

## Discussion

The present results show that the proportion of ZIKV-positive women was significantly greater in precarious women overall, and mostly in undocumented foreign women. Indeed, in the subgroup of precarious women, there was a trend suggesting greater exposure for undocumented women, presumably because they often live in shantytowns where vector proliferation and contact is greater than among documented precarious women who may have greater access to subsidized housing. Overall Western French Guiana was more affected than the rest of the territory[6]. The greater proportion of women with positive IgG and negative IgM serology without immunocapture whereas 9.82% of women with normal health insurance had positive IgG and negative IgM serology without immunocapture, P<0.00in the precarious group may also have reflected earlier infections during pregnancy, thus a proxy for greater risk.

Women living beyond the Coastal areas, where vector control is not as developed as in urban areas, were also significantly more likely to be infected by ZIKV.

The variables used are a coarse proxy for environmental and behavioral aspects linked to poverty, which were not measured. However they have been robustly associated with a number of chronic and acute health problems in French Guiana. [7,11]The present study was hospital-based and the exact health insurance status was not always recorded for different reasons: lack of time in a busy obstetrical ward, communication problems because of the very large number of women who do not speak French. Although, the type of health insurance is based on legal or illegal residence, length of stay, and on having an income below a certain threshold, it is conceivable that health insurance as a proxy for social precariousness is not perfect and that some women in the regular health insurance group may have been precarious. The limited number of adjustment variables, notably the living area used for the analysis did not allow to precisely study confounding between individual and collective socio-economic determinants. Despite these limitations, the present results were not a quest for any significant p value but were a clearly defined *a priori* hypothesis that was tested with the available data from the 2 biggest maternities in French Guiana, which capture most deliveries.

**Table 1. Characteristics of the precarious and non-precarious women.**

|  | Precarious | | |
| --- | --- | --- | --- |
|  | **No** | **Yes** | **Total** |
| **Residence location** | | | |
| *Savanas* | 15 | 30 | 45 |
|  | 33.33 | 66.67 | 100.00 |
| *Western* | 405 | 1740 | 2145 |
|  | 18.88 | 81.12 | 100.00 |
| *Central Coastal* | 517 | 646 | 1163 |
|  | 44.45 | 55.55 | 100.00 |
| *Eastern* | 14 | 31 | 45 |
|  | 31.11 | 68.89 | 100.00 |
| **Age** | | | |
| *<20* | 98 | 444 | 542 |
|  | 18.08 | 81.92 | 100.00 |
| *[20–30[* | 430 | 1189 | 1619 |
|  | 26.56 | 73.44 | 100.00 |
| *0–40[* | 393 | 740 | 1133 |
|  | 34.69 | 65.31 | 100.00 |
| *>40* | 43 | 101 | 144 |
|  | 29.86 | 70.14 | 100.00 |
| **Trimester at diagnosis** | | | |
| *pre-fertilization* | 13 | 9 | 22 |
|  | 59.09 | 40.91 | 100.00 |
| *1st trimester* | 161 | 359 | 520 |
|  | 30.96 | 69.04 | 100.00 |
| *2nd trimester* | 245 | 683 | 928 |
|  | 26.40 | 73.60 | 100.00 |
| *3rd trimester* | 202 | 582 | 784 |
|  | 25.77 | 74.23 | 100.00 |
| *term* | 316 | 784 | 1100 |
|  | 28.73 | 71.27 | 100.00 |
| *post-conception* | 3 | 14 | 17 |
|  | 17.65 | 82.35 | 100.00 |
| **At least 1 positive IgM** | | | |
|  | 123 | 562 | 685 |
|  | 17.96 | 82.04 | 100.00 |
| **At least 1 positive IgG** | | | |
|  | 143 | 668 | 811 |
|  | 17.63 | 82.37 | 100.00 |
| **Positive PCR** | | | |
|  | 38 | 45 | 83 |
|  | 45.78 | 54.22 | 100.00 |

These results, previous observations on dengue and CHIKV, observations in patient with chronic diseases, in pregnant women, and in persons renouncing to health care all suggest that social inequalities of health often affect the same populations with poorer living conditions and reduced access to care and prevention.[4,12] The present results emphasize that population approaches for a range of selected problems may be more pertinent than an array of

**Table 2. Crude and adjusted prevalence ratios (Poisson regression) for ZIKV infection in pregnant women in French Guiana in 2016.**

| | Confirmed ZIKV (n = 906) | Negative ZIKV (n = 2533) | PR (95% CI) | p | aPR(95% CI) | p |
|---|---|---|---|---|---|---|
| **Precariousness (N = 3439)** | | | | | | |
| **Not precarious** | 178 (18.5%) | 786 (81.5%) | - | - | - | - |
| **PUMa*** | 361 (27.6%) | 949 (72.4%) | 1.5 [1.3; 1.8] | <0.0001 | 1.4 [1.1; 1.6] | 0.001 |
| **AME**** | 270 (31.2%) | 595 (68.8%) | 1.7 [1.4; 2.0] | <0.0001 | 1.5 [1.2; 1.8] | <0.0001 |
| **No insurance** | 97 (32.3% ) | 203 (67.7%) | 1.8 [1.4; 2.2] | <0.0001 | 1.6 [1.2; 2.0] | <0.0001 |
| **Residence Localization (N = 3439)** | | | | | | |
| **Savannas** | 23 (51.1%) | 22 (48.9%) | 1.5 [1.3–1.8] | <0.0001 | 2.7 [1.7–4.1] | 0.008 |
| **Western** | 650 (30.3%) | 1495 (69.7%) | 1.5 [1.4–1.7] | <0.0001 | 1.5 [1.3–1.8] | <0.0001 |
| **Central Coastal** | 212 (18.2%) | 951 (81.8%) | _ | | _ | |
| **Eastern** | 12 (26.7%) | 33 (73.3%) | 1.3 [1.0–1.7] | 0.02 | 1.4 [0.8–2.5] | 0.3 |
| **Visitors** | 9 (22.0%) | 32 (78.1%) | 1.6 [1.2–2.0] | <0.0001 | 1.1 [0.6–2.2] | 0.7 |
| **Age (years)** | | | - | | | |
| **<20** | 159(29.3) | 383(70.7) | | | | |
| **[20–30]** | 411(25.4) | 1208(74.6) | 0.8 [0.7–1.03] | 0.12 | 0.9 [0.8–1.1] | 0.6 |
| **[30–40]** | 294(25.9) | 839(74.1) | 0.9 [0.7–1.07] | 0.21 | 1 [0.8–1.2] | 0.8 |
| **>40** | 41(28.5) | 103(71.5) | 0.9 [0.7–1.3] | 0.86 | 1.05 [0.7–1.5] | 0.7 |

vertical social programs in different populations. Indeed, many community approaches are presently funded as thematic (HIV prevention and testing, addictions. . .) and community worker thus often only focus on a single theme hence missing opportunities to improve outcomes for other important health problems. Relationship between social aspects and seropositivity to ZIKV, to DENV ($p<10^{12}$)[13], and to CHIKV ($p<10^{15}$) [14] suggests a complex interplay between individual factors and ecological/environmental factors.[15] At the individual level, underprivileged populations often have lower levels of health literacy; they often live in shacks surrounded by mosquito-breeding grounds and little protection from vectors. At the ecological level, these individual situations add up, and in addition, these populations generally gather in informal habitats in areas that are more likely to have high vector densities, and to be less prioritized by municipal and regional services.[16] Further studies should aim precisely define the potential multilevel causal paths underpinning this statistical association: A better mapping of vector breeding sites in priority areas, specific knowledge attitudes and behavior studies in underprivileged populations, a survey of the geographic distribution of vector-control interventions, notably relative to underprivileged areas. Such data would allow devising interventions aiming at improving health literacy and empowering populations relative to the prevention of vector-borne diseases, and improving the reach of vector-control by specifically targeting such areas.

In conclusion, during the ZIKV epidemic in French Guiana, precarious pregnant women and women living in Western French Guiana were significantly more affected by ZIKV than non-precarious women and women living in Central coastal areas.

## Supporting information

**S1 Checklist. STROBE checklist.**
(DOC)

## Author Contributions

**Conceptualization:** Claude Flamand, Mathieu Nacher.

**Data curation:** Edouard Hallet, Camille Fritzell, Séverine Matheus.

**Formal analysis:** Edouard Hallet, Mathieu Nacher.

**Investigation:** Dominique Rousset, Séverine Matheus.

**Methodology:** Claude Flamand.

**Resources:** Maryvonne Dueymes, Balthazar Ntab.

**Supervision:** Claude Flamand, Mathieu Nacher.

**Validation:** Dominique Rousset, Maryvonne Dueymes, Mathieu Nacher.

**Writing – original draft:** Edouard Hallet, Mathieu Nacher.

**Writing – review & editing:** Edouard Hallet, Claude Flamand, Dominique Rousset, Timothée Bonifay, Camille Fritzell, Séverine Matheus, Balthazar Ntab, Mathieu Nacher.

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
