## [Decision Letter · Decision Letter 0]

28 Jan 2020

Dear Pr. Nacher,

Thank you very much for submitting your manuscript "ZIKV infection in pregnant women in French Guiana: more precarious-more at risk." for consideration at PLOS Neglected Tropical Diseases. As with all papers reviewed by the journal, your manuscript was reviewed by members of the editorial board and by several independent reviewers. In light of the reviews (below this email), we would like to invite the resubmission of a significantly-revised version that takes into account the reviewers' comments. 

Overall the reviews were favorable, however, we cannot make any decision about publication until we have seen the revised manuscript and your response to the reviewers' comments. Your revised manuscript is also likely to be sent to reviewers for further evaluation.

Sincerely,

William B Messer

Associate Editor

Scott Halstead

Deputy Editor

Reviewer's Responses to Questions

**Key Review Criteria Required for Acceptance?**

**Methods**

-Are the objectives of the study clearly articulated with a clear testable hypothesis stated?

-Is the study design appropriate to address the stated objectives?

-Is the population clearly described and appropriate for the hypothesis being tested?

-Is the sample size sufficient to ensure adequate power to address the hypothesis being tested?

-Were correct statistical analysis used to support conclusions?

-Are there concerns about ethical or regulatory requirements being met?

Reviewer #1: The objectives of the study are clear. The study design and the population have to be more precisely detailed. The sample size is large. Analysis is correct but lack go adjustment. No concern about ethical or regulatory requirements.

Reviewer #2: The objectives of this study are clearly stated in the the hypothesis "that in pregnant women Zika Virus (ZIKV) was more frequent in precarious women than in non-precarious women as reflected by their health insurance status". The analytic approach supports the aims of the hypothesis. 

However, the stated study design of "a retrospective case-control" is incorrect in reference to the study sample and analytic approach. The sample of of "all deliveries or medical abortions between January 2016 and December 2016" and the use of prevalence ratios suggest that a "cross-sectional study design" better describes this work. The sample was taken from a singular point in time. The authors then estimate the association between disease prevalence (ZIKV serology) and other prevalent health factors (precariousness via health insurance status) at that point in time. All of which is suggestion of a cross sectional study, not a case control.

Reviewer #3: How can we be sure that woment with regular health insurance are not precarious?

The reason why neutralization for IgG could not be implemented should be explained.

Why is there no mention of an ethical committee for this study?

**Results**

-Does the analysis presented match the analysis plan?

-Are the results clearly and completely presented?

-Are the figures (Tables, Images) of sufficient quality for clarity?

Reviewer #1: Analysis match the analysis plan but the results have to be completed to be comprehensive.

Reviewer #2: Overall, the analysis is clearly presented and the paper proceeds in a logical manner. However, the measure of association in the body of the paper, adjusted prevalence ratio (APR) does not match the measure in abstract (adjusted odds ratio, aOR). Please change the measure and language in the abstract (AOR, "associated with a higher risk of ZIKV") to match the body of the text (APR, "associated with a significant increase of ZIKV prevalence").

Reviewer #3: We need to understand who are the 837/6654 women who had positive IgG and negative IgM serology without immunocapture. It could represent a biais for the final analysis.

The 2,621 women (39%) with unknown health insurance lived principally in Kourou

municipality (20%). This is a high rate, the risk that a high number of these women are precarious should be discussed.

For abortion, please specify if there are spontaneous or not.

It is stated that among precarious, irregular immigrants seemed more exposed 97/300 (32.3%) than French nationals 361/ 1310 (27.6%) (p=0.10). It need to be discussed further.

**Conclusions**

-Are the conclusions supported by the data presented?

-Are the limitations of analysis clearly described?

-Do the authors discuss how these data can be helpful to advance our understanding of the topic under study?

-Is public health relevance addressed?

Reviewer #1: The conclusions are supported by the data. Limitations are clearly described but could be discussed further.

Reviewer #2: The conclusions are robust and link well to public health relevance in the context of the ZIKV epidemic in French Guiana. 

The second to last paragraph could benefit from some clarifications:

- What specific vertical social programs are less beneficial than which population health approaches, in the context of preventing/addressing infectious disease in French Guiana? 

- For the interplay between the individual factors of precariousness/poverty and the ecological/environmental factors resulting in the outcome of ZIKV as well as DENV and CHIKV: 

~What are potential hypotheses for the possible associations or pathways resulting infection? 

~How could these be verified and by what experimental designs? 

~How could these future studies be used to inform public health in a manner that would mitigate or prevent the next infectious outbreak, especially among a vulnerable/precarious/impoverished population?

Reviewer #3: The epidemiology and the repartion of ZIKV infection in FG should be discussed. We need to know where most of the cases occured in this territory.

**Editorial and Data Presentation Modifications?**

Reviewer #1: Add a table with description of women by study categories.

Reviewer #2: - Change the study design from case-control to cross-sectional.

- Change the measure of association in the abstract from AOR to APR, remove reference to the risk of ZIKV and add a reference to the prevalence of ZIKV.

- Clarify some of the language and phrasing in the discussion section. 

Please refer to the suggested minor modifications in the above sections for more details.

Reviewer #3: There is no author listed with affiliation 4 Regional epidemiology unit of French Public Health Agency, Cayenne, French Guiana.

**Summary and General Comments**

Reviewer #1: General comments

This study addresses an important topic in an area where data are limited: Zika virus infection in pregnant women in a French oversea territory located in the Amazon area and its relation to social insecurity. The proportion of women affected by Zika infection is very high and women in precarious situations seem more often concerned.

Although the study is monocentric, the number of pregnant women included is large. 

In view of the existence in France of health insurance coverage for the poor and the undocumented (Complémentaire Santé Solidaire -CSS- and Aide Médicale d’Etat – AME-), the status relating to health insurance is a good proxy for measuring precariousness. However, it could be useful to discuss the lack of more precise data on the social situation. In particular, how were people without health insurance coverage treated?

As the rights to CSS and AME run 12 months regardless of changes in administrative status, the social situation of people may be different than that announced (undocumented migrants, precarious and non-precarious French resident )

Specific comments for revision

Title:

Zika Virus should be written in full in the title

Abstract:

Avoid repetitions “Most pregnant women were precarious 2275/3439. Most women were precarious but…”

Please specify the number of women belonging to the 3 analysis categories (undocumented, precarious, non-precarious)

Avoid “as elsewhere,” in the conclusion.

In addition to the inequalities of vector control by neighborhoods, it would be interesting to question the differences in personal behavior in the application of these same measures

Manuscript:

Authors’ affiliations: an affiliation does not correspond to any author (4 Regional epidemiology unit of French Public Health Agency, Cayenne, French Guiana)

Detail abbreviations: GDP

Between the summary which speaks of a monocentric study and the results which speak of 3 sites, it is not clear who was included in this study

Corrected “assurance”.

There are inaccuracies to correct on French health insurance coverage

The Universal health coverage (=CMU) (basic health insurance coverage implemented in 1999) no longer exists since 2016, it was merged with national Health Insurance (Assurance maladie) during the Universal Health Protection reform (Protection universelle maladie -Puma-). In addition, the CMU does not have a 3-month residency obligation in France, unlike the AME.

The complementary universal health coverage (=CMUC) changed its name on 1/11/2019 when it was merged with Aid to complementary health (=ACS) and is now called complementary health solidarity (Complémentaire Santé Solidaire = CSS)

In addition, the CMU does not have a 3-month residency obligation in France, unlike the AME

Please specify for CSS and AME the resource criteria which explain why they can be used as a proxy for their social status

Prefer the term national Health Insurance (Assurance maladie) to that of social security (Sécurité Sociale) for others

The paragraph on the biological diagnosis must be reviewed and clarified

Avoid “ extraordinary context”

The methods of collecting data must be more precisely described. Likewise, it is not clear which motherhood participated and why.

A flow chart specifying the site for monitoring women and their health coverage would be welcome

If prevalence ratio from Poisson regression were used, why in the abstract we found Odds Ratio?

The procedures for diagnosing Zika infection should be described much more precisely:% of women diagnosed with PCR,% of women diagnosed with serology and in what term, with what result of the previous serology, etc. A detailed table would be welcome. These results will ideally be presented according to the 3 study categories

“and no insurance: APR = 1.59”? You did not specify before having included people without health insurance. What are they? Profile?

Table 1: detail the abbreviations at the bottom of the table. Specify the method of analysis in the title.

Why not have adjusted model on age? Other adjustment variables, if available, would be welcome. The model as it is is difficult to interpret.

Discussion:

Can you discuss further why vector control measures are less implemented in certain regions and the policy of the regional health agency on this point. In particular, a qualitative insight into the real situation in the districts concerned would be useful for understanding the results.

The discussion could be enriched with international references having investigated the relationships between arboviruses and social situation

Reviewer #2: This is an interesting analysis acceptable for publication following minor revisions. One minor thematic point relates on the authors comments on "The unusual distribution of mircocephaly in Brazil, suggested a possible link between poverty and the risk of ZIKV related microcephaly..." The association between poverty and increased risk of infectious disease should not be considered unusual. Especially given the authors prior notes concerning the lack of sanitation and presence of disease carring insects in poor neighborhoods in French Guiana.

Reviewer #3: This an interesting study that highlights the fact that precarious people are more at risk of Zika infection in French Guiana. We need to better understand the methodology and the recommandation for public Health.

PLOS authors have the option to publish the peer review history of their article (what does this mean?). If published, this will include your full peer review and any attached files.

Reviewer #1: Yes: Nicolas Vignier

Reviewer #2: No

Reviewer #3: No
---

## [Editor Report · Decision Letter 1]

27 Feb 2020

Dear Pr. Nacher,

Thank you very much for submitting your manuscript "ZIKA Virus infection in pregnant women in French Guiana: more precarious-more at risk." for consideration at PLOS Neglected Tropical Diseases. As with all papers reviewed by the journal, your manuscript was reviewed by members of the editorial board and by several independent reviewers. The reviewers appreciated the attention to an important topic. Based on the reviews, we are likely to accept this manuscript for publication, providing that you modify the manuscript according to the review recommendations. 

Please review the minor edits and one question from the Associate Editor in the uploaded version of the tracked-changes manuscript.

Sincerely,

William B Messer

Associate Editor

Scott Halstead

Deputy Editor

Please review the minor edits and one question from the Associate Editor in the uploaded version of the tracked-changes manuscript.
---

## [Editor Report · Decision Letter 2]

3 Mar 2020

Dear Pr. Nacher,

We are pleased to inform you that your manuscript 'ZIKA Virus infection in pregnant women in French Guiana: more precarious-more at risk.' has been provisionally accepted for publication in PLOS Neglected Tropical Diseases.

Before your manuscript can be formally accepted you will need to complete some formatting changes, which you will receive in a follow up email. A member of our team will be in touch within two working days with a set of requests.

Best regards,

William B Messer

Associate Editor

Scott Halstead

Deputy Editor

---

## [Editor Report · Acceptance letter]

17 Mar 2020

Dear Pr. Nacher,

We are delighted to inform you that your manuscript, "ZIKA Virus infection in pregnant women in French Guiana: more precarious-more at risk.," has been formally accepted for publication in PLOS Neglected Tropical Diseases.

Best regards,

Serap Aksoy

Editor-in-Chief

Shaden Kamhawi

Editor-in-Chief
